# Reassessing the distribution of *Burkholderia pseudomallei* outside known endemic areas using animal serological screening combined with environmental surveys: The case of Les Saintes (Guadeloupe) and French Guiana

**Mégane Gasqué**[1,2], **Vanina Guernier-Cambert**[2,3]*, **Gil Manuel**[4], **Rachid Aaziz**[1], **Jules Terret**[1], **Thomas Deshayes**[1], **Xavier Baudrimont**[5], **Sébastien Breurec**[6], **Emma Rochelle-Newall**[2], **Karine Laroucau**[1]

**1** Animal health laboratory, Bacterial Zoonosis Laboratory, Anses, Paris, France, **2** Sorbonne Université, UPEC, IRD, CNRS, INRAE, Institute of Ecology and Environmental Sciences-Paris (iEES-Paris), Paris, France, **3** Faculty of Veterinary Technology, Kasetsart University, Bangkok, Thailand, **4** Veterinary clinic of Carmel, Guadeloupe, France, **5** Unit for Animal and Plant Health and Protection Environment (SPAVE), Directorate for Environment, Agriculture, Food and Forestry (DEAAF), Veterinary and Phytosanitary Inspection Service (SIVEP), French Guiana, France, **6** Pasteur Institute of Guadeloupe, Morne Jolivière, Guadeloupe, France

* vanina.guernier@ird.fr

## Abstract

### Background

Melioidosis, an emerging infectious disease that affects both humans and animals, is caused by the soil-dwelling bacterium *Burkholderia pseudomallei*. It is endemic in South and Southeast Asia, and northern Australia, causing an estimated 165,000 human cases annually worldwide. Human cases have been reported in the French West Indies (Martinique and Guadeloupe) since the 1990s. Conversely, no human cases have been reported in French Guiana, a French territory in South America. Our study aimed to investigate whether *B. pseudomallei* is locally established in Guadeloupe and French Guiana using animals as a proxy.

### Methodology/principal findings

Blood samples were collected from different animals from 56 farms in French Guiana (n = 670) and from two goat farms in Les Saintes (n = 31), *part of the Guadeloupe archipelago* and tested by enzyme-linked immunosorbent assay (ELISA). In Les Saintes, a serological follow-up was performed, and soil, water and goat rectal swabs were collected and analyzed by culture and PCR. The highest seroprevalence rates (39%) were observed in goats in Les Saintes, followed by horses (24%) and cattle (16%) in French Guiana. In the two goat farms, supplementary analyses detected *B. pseudomallei* from one goat rectal swab, and a *B. pseudomallei* strain was isolated from the soil.

**Data Availability Statement:** All relevant data are within the manuscript and its Supporting Information files.

**Funding:** This study was supported by the Ile de France Region (Dim1Health) (ERN, MG) and the European Commission Directorate-General for Health and Consumers (EC no.180/2008) (KL). We thank the French Institut de Recherche pour le Développement (IRD) and the French Agence nationale de sécurité sanitaire de l'alimentation, de l'environnement et du travail (ANSES) for their financial support. The funders had no role in the design of the study, collection and analysis of the data, decision to publish, or preparation of the manuscript.

**Competing interests:** The authors have declared that no competing interests exist.

## Conclusions/significance

Our animal serological data suggest the presence of *B. pseudomallei* in Les Saintes and French Guiana. In Les Saintes, environmental surveys confirmed the endemicity of the bacteria, which is consistent with documented human cases of melioidosis on the island. We did not conduct an environmental survey in French Guiana. Nevertheless, our serological results call for local environmental surveys and a retrospective reassessment of human infections with melioidosis-like symptoms.

## Author summary

Melioidosis, a disease caused by the environmental bacterium *Burkholderia pseudomallei*, has historically been described in Southeast Asia and northern Australia. However, recent studies have demonstrated its presence outside these areas, both in the environment and in patients without a history of travel to known endemic areas. In addition, the predicted increase in extreme climatic events in the near future could increase the prevalence of the disease and lead to its emergence in new areas. For these reasons, it is important to identify areas at risk outside of known endemic areas.

Hypothesizing that we would have little chance of finding *B. pseudomallei* through random environmental surveys, we used serological testing to find evidence of past exposure to the bacteria in apparently healthy domestic animals. We identified seropositive animals in Les Saintes and French Guiana. We then searched for the presence of *B. pseudomallei* in the immediate vicinity of the seropositive animals in Les Saintes, and isolated it in the soil of a goat farm. Our study suggests that domestic animals could be used as sentinels for the detection of melioidosis outside of countries with frequent human cases.

## Introduction

Melioidosis is an opportunistic infectious disease with a significant prevalence in South Asia, Southeast Asia, and northern Australia, primarily due to environmental exposure to *Burkholderia pseudomallei*. This soil-dwelling bacterium can affect both humans and animals through inhalation, ingestion, or skin contact [1]. *B. pseudomallei* is listed as a Class 1 pathogen by the US Centers for Disease Control and Prevention (CDC) and as a selected agent under the Highly Pathogenic Microorganisms and Toxins (MOT) regulation by the French National Agency for the Safety of Medicines and Health Products (ANSM).

The global burden of melioidosis was predicted to be 165,000 human cases in 2015 [2]. In addition, reports of human and, occasionally, animal cases are increasing along with environmental evidence of the bacteria worldwide, highlighting the likely presence of the bacterium on all continents, including Africa [3,4], North America [5,6], and South and Central America [7–9].

The clinical presentation of the disease is highly variable between individuals and species, ranging from chronic forms to acute septicemia [1]. Diagnosis and treatment of human melioidosis are complicated by the lack of pathognomonic clinical features and the antibiotic resistance of the bacterium to several antibiotics. Risk factors such as diabetes and chronic diseases contribute to increased susceptibility [10]. In animals, melioidosis affects a variety of species with multiple cases documented in domestic animals such as cats, cattle, deer, dogs, goats,

equids, sheep, and sporadic cases in other species, including marine mammals, birds and even crocodiles in zoos or wildlife parks [11].

While accurate epidemiological data and diagnostic tools are lacking for human melioidosis, they are even more limited for animal melioidosis. Diagnostic approaches for animal melioidosis are generally the same as for humans [11–13]. Although diagnostic methods such as indirect hemagglutination (IHA), complement fixation test (CFT) and enzyme-linked immunosorbent assay (ELISA) have been developed for commonly tested species such as small ruminants, pigs, and equids [14–16], standardized tests for the diagnosis of melioidosis in all susceptible species are currently lacking. A recent study demonstrated the effectiveness of serological tests developed for glanders (a disease caused by *B. mallei*, a bacterium closely related to *B. pseudomallei*) in accurately detecting melioidosis in equids [17]. These tests include both a reference method (CFT) and alternative methods (ELISA, Western blot).

The bacterium *B. pseudomallei* is mainly found in moist clay soils and in turbid water [18,19]. Its distribution in soils is strongly influenced by climatic events, with more cases observed during the rainy season and sporadic contamination peaking after extreme weather events such as storms or floods [20,21]. Environmental detection of *B. pseudomallei* is challenging and specific culture protocols are required to isolate the bacterium [22]. Detection may be more difficult in areas of low endemicity or emergence, where the bacterium may be present at lower concentrations and/or in limited locations. A new culture medium using erythritol as a carbon source has recently been developed to improve culture isolation from soil samples [23]. It was successfully used to isolate *B. pseudomallei* from a rice farm in south central Ghana, West Africa [24].

Cases of melioidosis have been reported in both South, Central, and North America. This includes the French West Indies, i.e., Guadeloupe and Martinique, where 21 human cases have been reported since 1993 [25–30], some of whom had no history of travel outside of their island of residence. In contrast, no human cases have been reported in French Guiana (Guiana Shield, South America), while human cases have been reported in neighboring Northeastern Brazil [9].

The first objective of this study was to assess the exposure of different livestock species (goats, cattle, horses, sheep) to *B. pseudomallei* in Les Saintes, a group of islands in the Guadeloupe archipelago, and in French Guiana. The second objective was to confirm the local presence of the bacteria in farms with seropositive animals by molecular testing of fecal and environmental samples, and isolation of *B. pseudomallei* by culture.

## Materials and methods

### Ethics statement

Animal sampling protocols have been approved by the Institute of Research and Development (IRD) Ethics Committee (April 21, 2022). The application submitted to the Ethics Committee covers blood and swab collection by veterinarians on domestic animals in Les Saintes and French Guiana for the period January 1, 2021 to December 31, 2023. Written consent was also obtained from the two goat breeders in Les Saintes. In addition, a Nagoya Protocol dossier (European level, National Decree No. 2017–848 of May 9, 2017) was approved for animal and environmental samplings. Anses is authorized to handle *B. pseudomallei* in accordance with the ANSM regulation on MOT, and to import, analyze and store environmental samples in accordance with the EU Regulation 2019–829 (DAAF, Prefectural Order of 25 August 2022).

### Initial animal sampling in French Guiana and Les Saintes

A total of 56 farms were sampled in French Guiana, mainly along the north coast where 80% of the population lives (Fig 1B). These farms included both single-species and multi-species

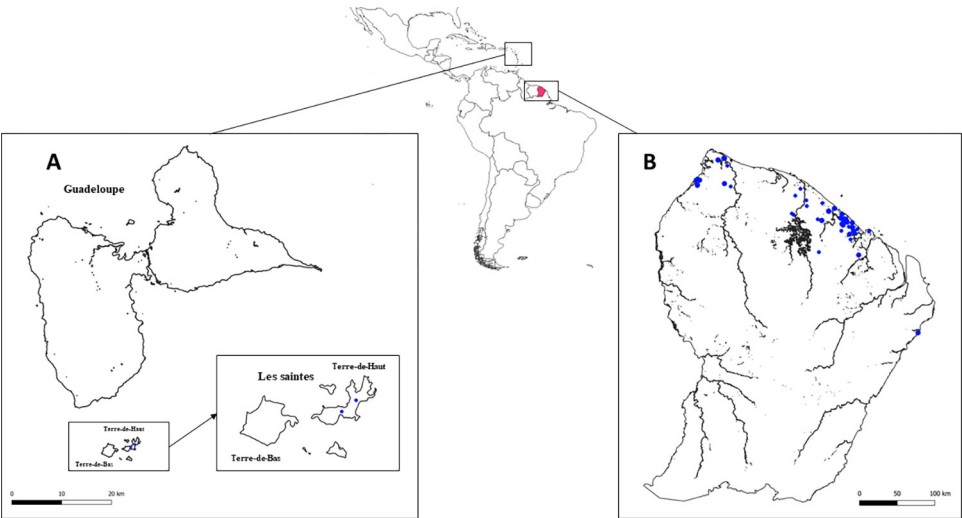

**Fig 1. Map of sampling sites.** Blue dots represent farms where animals were tested in (A) Les Saintes (n = 2) and (B) French Guiana (n = 56). Les Saintes is part of the Guadeloupe archipelago, and consists of nine islands, of which only the two largest are inhabited, Terre-De-Haut and Terre-De-Bas; only Terre-De-Haut was sampled. The map layers (country boundaries) of South America, Martinique and French Guiana were downloaded from the geoBoundaries website https://www.geoBoundaries.org/simplifiedDownloads.html).

farms, ranging from 2 to 31 animals per farm. Among the different regions of French Guiana, *18 farms were located in Macouria, 11 in Kourou, 7 in Mana, 7 in Saint Laurent du Maroni, 4 in Sinnamary, 3 in Montsinery Tonnegrande, 2 in Cayenne, and 1 in Iracoubo, Matoury Roura, and Saint George de l'Oyapock*. Notably, 10 of these farms had more than one species, including cattle, sheep and horses. Blood samples were collected between March 2021 and September 2023 as part of an annual surveillance program conducted by the Direction of Food, agriculture and forests (DAAF—Direction de l'Alimentation, de l'Agriculture et de la Forêt) in French Guiana. A subsample of this serobank was provided by the DAAF veterinarian and included (n = 670): 361 cattle from 37 farms, 131 sheep from 15 farms, 100 goats from 10 farms, 63 equines from six farms, and 15 pigs from one farm.

In Les Saintes, on the island of Terre-De-Haut, two goat farms were included in this study and monitored by a local veterinarian: Farm A housed approximately 40 goats and Farm B housed approximately 60 goats. Sampling was not exhaustive as they were free roaming (Fig 1A). On both farms, mixed-sex goats grazed freely in open pastures, consuming local vegetation, and no birth control measures were used, hence the approximate herd sizes reported by the farmers. The topography of the pastures was characterized by steep terrain and both were susceptible to significant surface water runoff. In November 2021, 31 goats were sampled for blood: 14 from Farm A and 17 from Farm B. After blood clotting and centrifugation at 900 rpm for 10 min, sera were collected, heat-inactivated at 60˚C for 30 min and stored at -20˚C until analysis. These two farms were selected for further investigation based on their accessibility and on previous reports of human cases in Les Saintes.

## Serological analysis

**GLANDA ELISA.**   The ID ScreenGlanders Double Antigen Multi-species ELISA test (Innovative Diagnostics, Grabels, France) (hereafter referred to as GLANDA ELISA) was used according to the manufacturer's instructions. This test does not rely on the use of a specific conjugate and can therefore be used for multiple animal species. The relative amounts of antibodies in the serum samples were calculated with reference to the positive and the negative

controls provided in the test. The serological titer (expressed as the sample to positive ratio or S/P%) was calculated from the optical density (OD) reading at 450 nm using the modified formula: S/P% = ((OD sample—OD negative control) / (OD positive control—OD negative control)) x 100. Serum samples with S/P% <70% were considered negative and ≥70% were considered positive.

The GLANDA ELISA test was originally developed for the diagnosis of glanders. A *B. pseudomallei* seropositivity result using this commercial test is therefore considered as presumptive rather than absolute; all seropositivity rates in our study were calculated based on these presumptive positive numbers.

**Additional serological tests on equine sera, French Guiana.**  The complement fixation test (CFT), the reference test for equine glanders, was performed as previously described using the cold method with the Malleus CFT antigen and the associated positive and negative controls (Bioveta, Czech Republic) [31]. Serum samples were initially tested at 1/5 dilution. Samples with 100% hemolysis were considered negative, those with 25–75% hemolysis were considered equivocal, and those with 100% inhibition of hemolysis were considered positive. All samples initially identified as equivocal or positive were retested over a range of dilutions. The "titer" is a six-digit barcode corresponding to the intensity of hemolysis inhibition (0 = 0%; 1 = 25%; 2 = 50%; 3 = 75%; 4 = 100%) at the reciprocal dilutions (1/5, 1/10, 1/20, 1/40, 1/80, 1/160) for each sample. Anti-complement activity (due to incomplete elimination of complement proteins during the serum heat inactivation) was checked for each serum.

In addition, the Luminex assay, originally developed for the diagnosis of equine glanders, was performed as previously described using the heat shock protein (GroEL) (BPSL2697) and hemolysin-coregulated protein (Hcp1) (BPSS1498) antigens [32]. A serum from a horse naturally infected with *B. mallei* (identified as MRI#1) was used as a positive control to determine S/P%, which was calculated for each antigen using the same formula as for GLANDA ELISA. S/P% values greater than 45% for GroEL and 43% for Hcp1 were considered positive, as previously established [31].

**Follow-up sampling of selected GLANDA ELISA positive farms.**  Additional serological monitoring was carried out on selected farms with GLANDA ELISA positive animals. In French Guiana, additional serological testing was carried out on a horse farm in January 2022 and a more comprehensive longitudinal study was conducted on two goat farms in Les Saintes.

Regarding the longitudinal study in Les Saintes, after the initial sampling in November 2021 (described above), blood samples and rectal swabs or feces were collected by the local veterinarian from April 2022 to December 2022 on farm B, and from April 2022 to March 2023 on farm A (Fig 2). Due to the free-range nature of the farms, goat capture was not exhaustive during each field visit. In April 2022, as many animals as possible were captured, sampled and identified with a plastic collar. However, due to the loss of these plastic collars, microchip re-identification was conducted in August 2022. This corresponds to sampling #1 of the longitudinal study (sampling #2 occurred in December 2022, sampling #3 in March 2023; see Fig 2). Blood samples were processed identically to the initial visit, and rectal swabs were stored dry at -20˚C until PCR analysis.

## Environmental sampling at the two goat farms in Les Saintes

Soil and water samples were collected from both goat pastures (farms A and B) in December 2022, i.e., during the rainy season (Fig 2).

## Soil samples

Soil samples (n = 50 in farm A, n = 42 in farm B) were collected along transects perpendicular to the main slope of the pasture. Transect lines were at least 6 m apart and sampling points

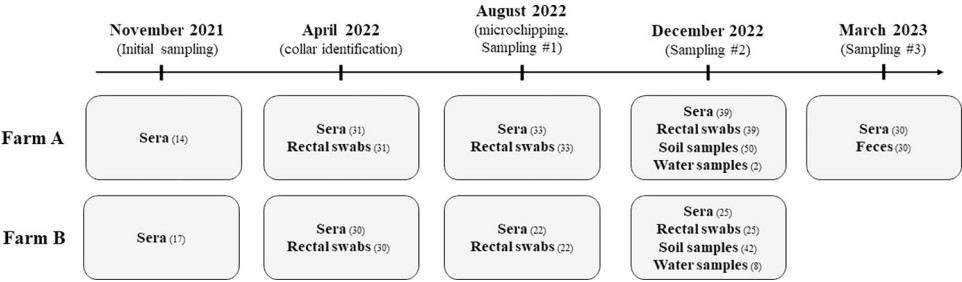

**Fig 2. Timeline of sampling at the two goat farms in Les Saintes (November 2021—March 2023).** November 2021 corresponds to the initial sampling. All other samplings correspond to the follow up study. The longitudinal studies, multiple sampling of the same individuals, occurred between August 2022 and March 2023 for farm A, and between August 2022 and December 2022 for farm B, after the animals were microchipped. The numbers in parentheses correspond to the number of samples collected from captured animals and the environment, on farms A and B.

along the same transect line were 5 m apart. Samples were taken at a depth of 30 cm using a standard soil auger, which was disinfected with absolute ethanol after each sampling. Approximately 200 g of soil was collected and placed in sealed Ziploc bags to preserve moisture. Samples were stored in the dark and at room temperature (approximately 25°C) until shipment to France. Samples were analyzed within two weeks of collection. Each soil sample was homogenized manually and 10 g of soil was subsampled and added to 10 mL of sterile water. After vigorous vortexing, the suspension was shaken on an orbital shaker at 160 rpm for 1 h and then allowed to settle for 10 min. A 1 mL sample of supernatant was centrifuged at 8,000 rpm for 10 min and the resulting pellet was stored at -80°C until PCR analysis. A further subsample of supernatant was used for a two-step enrichment. First, 1 mL of supernatant was mixed with 9 mL of Ashdown broth [33] and incubated at 37°C. After 48 h, the Ashdown broth was mixed and 1 mL was transferred to 9 mL of erythritol broth [23] and incubated at 37°C. After 96 h, 1 mL of erythritol broth was centrifuged at 8,000 rpm for 10 min and the resulting pellet was stored at -80°C until PCR analysis. The remaining 9 mL of broth was stored at 4°C.

## Water samples

Water samples were collected from goat drinking troughs on farms A (n = 2) and B (n = 8). Within two hours of collection, 50 mL of each water sample were filtered sequentially through a 3 μm filter followed by a 0.2 μm filter. All filters were folded in four, preserved in Lysogeny Broth (LB) (10 g peptone, 5 g yeast extract, 10 g NaCl for 1 L)-glycerol 20% and stored at -80°C until analysis. For analysis, the LB-glycerol 20% was discarded and the 0.2 μm filters were aseptically cut in half. One half was placed in 10 mL of Ashdown broth and followed the enrichment protocol described above, while the other half was stored at -80°C for PCR analysis.

## DNA extraction and PCR analysis (environmental and animal samples)

Soil supernatants without pre-enrichment were extracted using the DNeasy PowerSoil Pro kit (QIAGEN, Hilden, Germany), while supernatants from enriched soil samples were extracted using the High Pure PCR Template Preparation kit (Roche, Mannheim, Germany) according to the manufacturer's instructions. The heads of the rectal swabs were cut and placed directly into the lysis buffer and extracted using the High Pure PCR Template Preparation kit. The supernatants from the enriched water samples were extracted using the same kit. The half 0.2 μm filters used to filter the water samples were ribolyzed for three cycles of 20 s at 5,500 rpm (Lysing Matrix E two mL, MPBio, Eschwege, Germany) and DNA extraction was

performed on 800 μL of the supernatant using the High Pure PCR Template Preparation kit. Fecal samples were extracted using the following protocol: 2 g of feces were weighed and placed in 20 mL sterile water, then vortexed and placed under rotary agitation for 30 minutes. After settling for 15 minutes, the supernatant was collected and subjected to double enrichment as for the soil and water samples. DNA extraction was performed as for the soil and water samples. An internal inhibition extraction control (Xeno DNA control or Diagenode IPC, Thermofisher) was added to each sample.

Initial PCR screening was performed using an *aro*A-based real-time PCR assay [34] targeting the *B. pseudomallei* complex, which includes *B. pseudomallei*, *B. thailandensis*, as well as 6 other *Burkholderia* species [35]. DNA samples that tested positive were further tested using real-time PCR assays targeting specific regions of the *B. pseudomallei* genome (*orf*11, BPSS0087, BPSS0745, and/or BPTT4176-4290) [36,37] and/or *B. thailandensis* (70 kDa) [38]. Each PCR reaction contained 5 μL of DNA and 15 μL of reaction mix, 1X of Universal Mastermix TaqMan Fast Advanced 2X (Applied Biosystems, Vilnius, Lithuania), 0.5 μM of each primer, 0.1 μM of probe, 1X of IPC Diagenode and 2.8 μL of sterile water. The amplification procedure included an incubation at 50˚C for 2 min, followed by a denaturation step at 95˚C for 20 sec, then 45 cycles at 95˚C for 3 sec and 60˚C for 30 sec.

## Culture and screening of suspect colonies

Soil enrichment broths stored at 4˚C were homogenized and 100 μL were plated onto both conventional Ashdown agar and modified *B. cepacia* CHROMagar agar (BK992 CHROMagar, Paris, France) supplemented with 4% glycerol, 500 mg/mL gentamicin and 130 mg/mL fosfomycin (CHR_GGF). Plates were incubated at 37˚C and monitored daily. Suspect colonies (purple on Ashdown, green on CHR_GGF) were streaked on non-selective blood agar plates containing 5% horse serum (BA) and incubated at 37˚C for 24 h to obtain a pure culture. A loopful of colonies was resuspended in sterile water, heated at 100˚C for 20 min to break the cell membranes, and then centrifuged at 13,000 g for 10 min to pellet cell debris. The supernatant, rich in bacterial DNA, was used for PCR analysis (*aro*A and *orf*11 systems) to confirm colony identification. Isolates confirmed as *B. pseudomallei* by PCR were further characterized and their antimicrobial resistance patterns were identified.

## *B. pseudomallei* strain characterization

**Antibiogram.**   Antibiotic susceptibility testing was performed using the disk diffusion method according to the guidelines of the European Committee on Antimicrobial Susceptibility Testing (EUCAST) [39]. The antibiotics tested included: amoxicillin-clavulanate (20–10 μg), meropenem (10 μg), trimethoprim-sulfamethoxazole (1.25–23.75 μg), chloramphenicol (30 μg), imipenem (10 μg), ceftazidime (10 μg), and tetracycline (30 μg) (Bio-Rad, France), as per standard testing of human samples. In addition, we tested susceptibility to colistin (10 μg) and gentamycin (10 μg) (Bio-Rad, France), which were used as selective antibiotics in our culture media. Fresh 0.5 McFarland suspensions were prepared and plated on Mueller-Hinton agar (Bio-Rad, France). Antibiotic discs were then placed on the agar and plates were incubated at 35˚C for 18 h. EUCAST breakpoints were used for interpretation, except for colistin and gentamycin for which no EUCAST breakpoints have been published. For these two antibiotics the results are qualitative. As recommended by EUCAST, two reference strains, *Escherichia coli* ATCC 25922 and *Staphylococcus aureus* ATCC 29213, were included in the assay.

**Biochemical tests.**   The API 20 NE system (Biomérieux, France) was used according to the manufacturer's instructions. Bacterial suspensions were adjusted to a 0.5 McFarland standard and the API cassette was then incubated at 29˚C for 24 h. The web-based APIWEB

database (accessed 7 April 2023, Biomérieux, France, https://apiweb.biomerieux.com/) was used to interpret the profiles obtained. An oxidase test (oxidase reagent 50 x 0.75 mL, Biomérieux, France) was also performed on suspected colonies.

**Multilocus sequence typing (MLST).** Isolated strains were typed using MLST on 7 genes as described previously [40]. Amplified fragments were Sanger sequenced by Eurofins (Germany). Sequences were compared to the *Burkholderia pseudomallei* PubMLST database for MLST profiling (https://pubmlst.org/https://pubmlst.org/organisms/burkholderia-pseudomallei, accessed 17 February 2023) [41]. All published *B. pseudomallei* strains from South America were extracted from PubMLST: 19 were isolated from humans, 2 from animals and 2 from soil. Using BioNumerics software 7.6.3., we built a phylogenetic tree comparing the MLST profile of our strains with the 23 regional strains. The tree was built based on the 7 concatenated MLST genes and is a maximum parsimony tree with bootstrapping (n = 100).

## Results

### Serological survey in French Guiana and in Les Saintes

As stated in the methods, the GLANDA ELISA was originally designed for glanders diagnosis, therefore seropositivity against *B. pseudomallei* should be considered as presumptive.

In French Guiana, 16% (58/361) of cattle, 2% (2/100) of goats, 3% (4/131) of sheep, 24% (15/63) of horses and 0% (0/15) of pigs were seropositive (Table 1). The highest S/P% was observed in cattle, sheep and horses.

In Les Saintes, 39% (12/31) of goats were seropositive (Table 1). They were distributed over two farms (5/14 in farm A; 7/17 in farm B (S1 Table)), with S/P% ranging from 0 to 503%.

In French Guiana, 25 out of 56 farms tested positive, of which 10 were multi-species (Fig 3). Sixteen of the farms with positive tests were located in three communes: Macouria (n = 8), Kourou (n = 4), Sinnamary (n = 4). Interestingly, on positive farms, not all animals were positive and of the 25 GLANDA ELISA positive farms in French Guiana, 10 had animals with S/P% > 200.

### Additional analysis of horses from French Guiana and follow-up of farm #24

In French Guiana, horses originating from six out of 56 farms (2 to 31 horses per farm; n = 63) were further tested by two additional methods developed for the diagnosis of glanders: the reference CFT method [16] and the newly developed Luminex method using GroEL and Hcp1 antigens [32]. All horses with positive GLANDA ELISA results were confirmed positive for Hcp1 by Luminex, but negative for CFT and GroEL by Luminex, with the exception of farm 24 where 4 horses tested positive for CFT (**Tables 2 and S1**).

**Table 1. Summary of serological results (GLANDA ELISA) in domestic animals in French Guiana (n = 670) and in Les Saintes (n = 31).**

| Sampling site | French Guiana | | | | | Les Saintes |
|---|---|---|---|---|---|---|
| Animal species | Cattle | Goats | Sheep | Horses | Pigs | Goats |
| Number of positive animals/total (%) | 58/361 (16) | 2/100 (2) | 4/131 (3) | 15/63 (24) | 0/15 (0) | 12/31 (39) |
| [Min—Max] S/P% | [0–507] | [0–99] | [0–413] | [0–435] | [0–22] | [0–503] |
| Number of positive farms/total (%) | 17/37 (45) | 2/10 (20) | 2/15 (13) | 4/6 (66) | 0/1 (0) | 2/2 (100) |

S/P%: Serological titer expressed as the ratio of the sample to the positive control. Serum samples with S/P% ≥70% were considered positive, as per manufacturers recommendations for glanders diagnosis. Farms with at least one animal positive to the GLANDA ELISA test were considered positive.

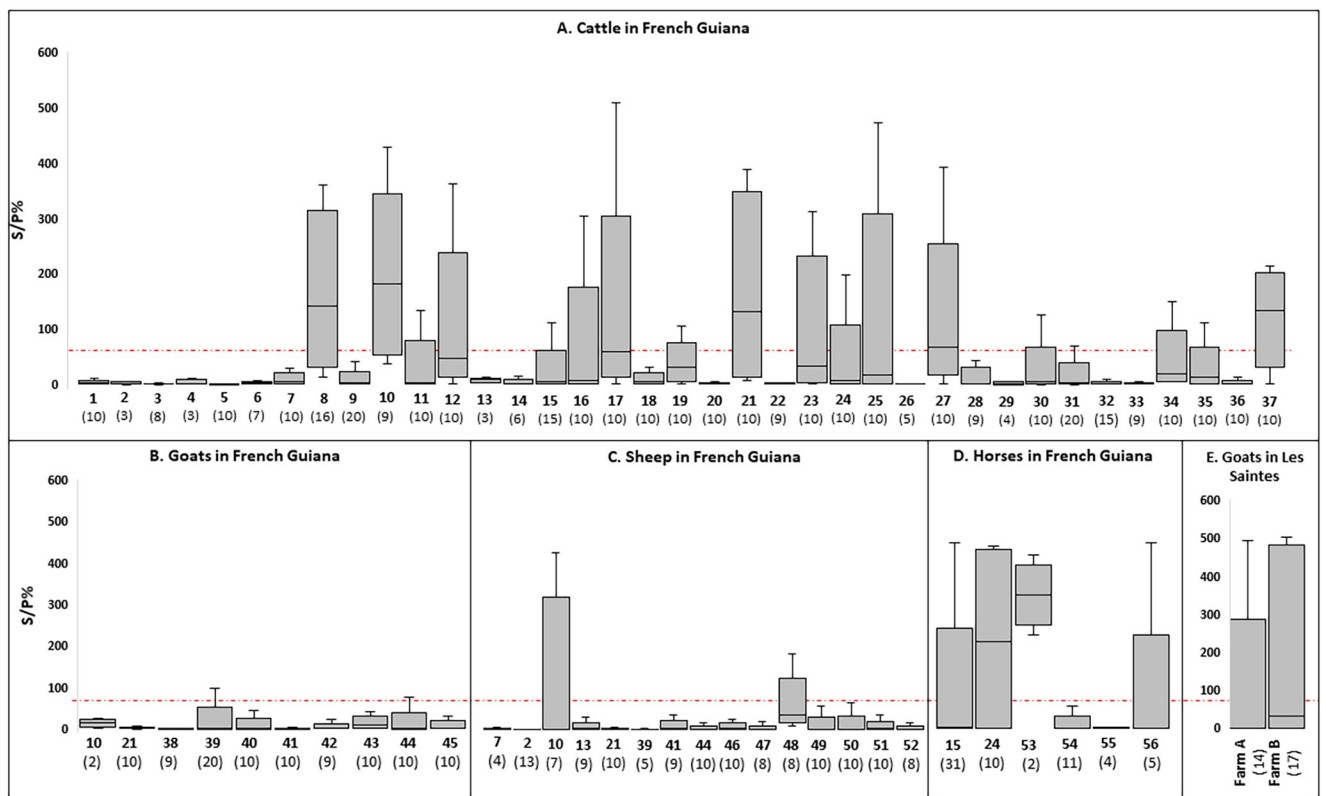

**Fig 3. Summary of GLANDA ELISA results per farm and species in French Guiana (A: cattle (n = 37), B: goats (n = 10), C: sheep (n = 15), and D: horses (n = 6)) and in Les Saintes (E: goats (n = 2)).** S/P%: Serological titer expressed as the ratio of the sample to the positive control. The x-axis of the graph represents farm identification numbers ranging from 1 to 56. Each farm is assigned a unique number, and if multiple species were present on the same farm, the same number is used. The total number of animals tested per farm is shown in parentheses below each farm number. The red dotted line represents the S/P% threshold (70%). The bar above and below the box plot indicates the minimum and maximum S/P%. The top and bottom of the box indicate the 1st and 3rd quartiles of S/P%. The bars inside the box indicate the median. Due to only one pig farm being tested, we have excluded these data from the figure.

## Focused study on two goat farms in Les Saintes

**Serological and PCR analysis.** To study the temporal evolution of the serological response in goats that initially tested ELISA-positive in November 2021, additional blood

**Table 2. Serological results of 63 horses from six farms in French Guiana.**

| Farm ID | Sampling date | Number of tested horses | GLANDA ELISA (P ≥ 70%) Positive horses | CFT Positive horses | Luminex—Hcp1 (P ≥ 43%) Positive horses | Luminex—GroEL (P ≥ 45%) Positive horses |
|---|---|---|---|---|---|---|
| 15 | March-21 | 31 | 7 | 0 | 9 | 1 |
| 24 | October-21 | 6 | 3 | 4 | 3 | 0 |
| | January-22 | 10 | 5 | 4 | 5 | 0 |
| 53 | March-21 | 2 | 2 | 0 | 2 | 0 |
| 54 | March-21 | 11 | 0 | 0 | 0 | 0 |
| 55 | March-21 | 4 | 0 | 0 | 0 | 0 |
| 56 | April-21 | 5 | 1 | 0 | 1 | 0 |

GLANDA ELISA: Enzyme-linked immunosorbent assay with the ID Screen Glanders Double Antigen Multi-species ELISA test (Innovative Diagnostics, Grabels, France). CFT: Complement fixation test. S/P%: Serological titer expressed as the ratio of the sample to the positive control. Luminex: bead-based assay targeting the recombinant proteins Hcp1 or GroEL. Thresholds used for each method are given in parentheses following the test name. Samples with anti-complementarity activity when tested by CFT (two horses on farm 15, three on farm 54 and one on farm 55) were considered as uninterpretable.

**Table 3. Serological and PCR results from two goat farms in Les Saintes.** The number of real-time PCR positive rectal swab samples relative to the total number tested is given for six PCR targets (one for the *B. pseudomallei* complex, four specific for *B. pseudomallei* and one specific for *B. thailandensis*). GLANDA ELISA: Enzyme-linked immunosorbent assay using the ID Screen Glanders Double Antigen Multi-species ELISA test (Innovative Diagnostics, Grabels, France). In November 2021, no rectal swabs were collected. * In March 2023, only fecal samples were collected for each goat.

| Farm | Sampling date | GLANDA-ELISA (sera) | | Real time PCR (rectal swabs) | | | | | | |
|---|---|---|---|---|---|---|---|---|---|---|
| | | | | Bpm complex | Bpm | | | | | Bt |
| | | pos/tot (%) | [min—max] %positivity | aroA | orf11 | BPSS087 | BPSS0745 | BPSS4208 | | 70 kDa |
| Farm A | November-21 | 5/14 (36) | [0–494] | not sampled | | | | | | |
| | April-22 | 4/31 (13) | [0–373] | 2/31 | 1/2 | 0/2 | 0/2 | 0/2 | | 0/2 |
| | August-22 | 8/33 (24) | [0–289] | 0/33 | 0/33 | na | na | na | | na |
| | December-22 | 11/39 (28) | [0–395] | 1/39 | 1/1 | 1/1 | 1/1 | 0/1 | | 0/1 |
| | March-23* | 7/30 (23) | [0–396] | 0/30 | 0/30 | na | na | na | | na |
| Farm B | November-21 | 7/17 (41) | [0–503] | not sampled | | | | | | |
| | April-22 | 7/30 (23) | [0–351] | 3/30 | 0/3 | 0/3 | 0/3 | 0/3 | | 0/3 |
| | August-22 | 5/22 (23) | [0–343] | 0/22 | 0/22 | na | na | na | | na |
| | December-22 | 7/25 (28) | [0–455] | 0/25 | 0/25 | na | na | na | | na |

Bpm: *Burkholderia pseudomallei*, Bt: *Burkholderia thailandensis*. na: not analysed (samples with *negative aroA* and *orf11 PCR analysis*). pos/tot (%): number of positive animals/total (percentage), S/P%: serological titer expressed as the ratio of the sample to the positive control.

samples were collected in April 2022, August 2022, December 2022 on both farms A and B in Les Saintes, and in March 2023 on farm A.

Seropositive animals were consistently detected at all time points on both farms, although the percentage of positive animals varied over time. It ranged from 13% to 36% in farm A and from 23% to 41% in farm B. The highest number of ELISA-positive animals was observed in both farms in November 2021 (Table 3).

Rectal swabs were collected in parallel with blood samples to test for the fecal shedding of *B. pseudomallei*. In total, six swabs tested positive with the *aro*A PCR targeting all species of the *B. pseudomallei* complex, of which two were confirmed positive for *B. pseudomallei*: one collected in April 2022 and one in December 2022 on farm A (Table 3).

On farm A, 20 goats were sampled three times, in August and December 2022 and March 2023; 13 were consistently GLANDA ELISA negative, five were consistently GLANDA ELISA positive and two changed status during the course of the 8-month study (**Tables 4 and S1**). Of note, six goats had S/P% greater than 200% on at least two sampling occasions. One goat with the lowest S/P% (≈150%) later tested negative in March 2023. Conversely, two goats with initially low S/P% later had higher values. Of the seven goats that were seropositive at different times, five were over one year old. The two identified mother-daughter pairs were consistently

**Table 4. Longitudinal serological study in two goat farms in Les Saintes.** GLANDA ELISA: Enzyme-linked immunosorbent assay using the ID Screen Glanders Double Antigen Multi-species ELISA test (Innovative Diagnostics, Grabels, France). S/P%: Serological titer expressed as the ratio of the sample to the positive control. Negative (S/P% <70); Positive (S/P% ≥70%); ns: not sampled (farm B was not sampled in March 2023).

| Farm | Number of animals | August 2022 | November 2022 | March 2023 |
|---|---|---|---|---|
| Farm A | 5 | Positive | Positive | Positive |
| | 1 | Positive | Positive | Negative |
| | 1 | Negative | Positive | Positive |
| | 13 | Negative | Negative | Negative |
| Farm B | 1 | Positive | Positive | ns |
| | 2 | Negative | Positive | |
| | 9 | Negative | Negative | |

negative. On farm B, 11 animals were sampled twice, in August and December 2022; eight were GLANDA ELISA negative twice, one was GLANDA ELISA positive twice, and two changed status during the course of the study (seroconversion). All three animals that were seropositive at least once were over one year old and had S/P% values greater than 200%.

## Environmental sampling in goat pastures in Les Saintes

In December 2022, 50 soil samples and two water samples (from the drinking troughs) were collected from farm A; and 42 soil samples and eight water samples (from the drinking troughs) were collected from farm B. All samples were analyzed by PCR and bacteriological culture. The 52 samples from farm A were all PCR negative, before and after enrichment. On farm B, two (22–10884_313 and 22–10884_336) out of 42 soil samples were PCR positive for the *B. pseudomallei* complex after enrichment, but only one sample (22–10884_313) was confirmed as *B. pseudomallei* with all four PCR specific targets (**Table 5**). All water samples were PCR negative.

The two PCR-positive soil enrichment broths (from soils 22–10884_313 and 22–10884_336) were plated on both Ashdown and modified CHROMagar *B. cepacia* agar media for isolation. Thirty suspect colonies from soil 22–10884_313 were PCR positive for the *B. pseudomallei* complex and all four *B. pseudomallei* specific targets, whereas the fifteen suspect colonies from soil 22–10884_336 were all PCR positive for the *B. pseudomallei* complex but negative for *B. pseudomallei* and *B. thailandensis*.

The *B. pseudomallei* colonies from soil 22–10884_313 initially appeared purple and smooth after 24 h at 37°C on Ashdown and then changed to a rough, dry texture after 48 h (Fig 4A). Similar morphological changes were observed on modified CHROMagar agar *B. cepacia*: colonies were green and smooth after 24 h of culture, but changed to a white, rough, and dry texture after 120 h (Fig 4B).

As all of the colonies obtained from plating soil 22–10884_313 looked similar, one *B. pseudomallei* colony (22–10884_313#20) from modified CHROMagar agar *B. cepacia* was selected for further analysis. The strain exhibited the API 20 NE profile 1-456-574 along with a positive result for the oxidase test. When compared with the profiles listed in APIWEB, the strain had the highest identification scores with *B. pseudomallei* (47.9%) and *B. cepacia* (45.3%). The antibiotic susceptibility profile of strain 22–10884_313#20 showed susceptibility to the tested antibiotics, all of which are commonly used in human melioidosis treatment (amoxicillin-clavulanate, ceftazidime, tetracycline, imipenem, meropenem, trimethoprim-sulfamethoxazole, chloramphenicol, colistin, and gentamycin) (**S2 Table**). The MLST sequence type of strain 22–10884_313#20 was ST92. In the *Burkholderia pseudomallei* PubMLST database,

**Table 5. Results of molecular testing of environmental samples from two goat farms in Les Saintes.** Soil and water samples collected in December 2022 were tested by real-time PCR for the presence of different *Burkholderia* spp after enrichment. Results are given as the ratio of positive samples to the total number of samples tested for each PCR target. na: not analyzed.

| Farm | Type of sample | Number of samples | Bpm complex | Bpm | | | | Bt |
|------|----------------|-------------------|-------------|------|---------|----------|----------|--------|
|      |                |                   | *aroA*      | *orf11* | BPSS087 | BPSS0745 | BPSS4208 | 70 kDa |
| A    | soil           | 50                | 0/50        | na   | na      | na       | na       | na     |
|      | water          | 2                 | 0/2         | na   | na      | na       | na       | na     |
| B    | soil           | 42                | 2/42        | 1/2  | 1/2     | 1/2      | 1/2      | 0/2    |
|      | water          | 8                 | 0/8         | na   | na      | na       | na       | na     |

*Bpm*: Burkholderia pseudomallei, Bt: Burkholderia thailandensis.

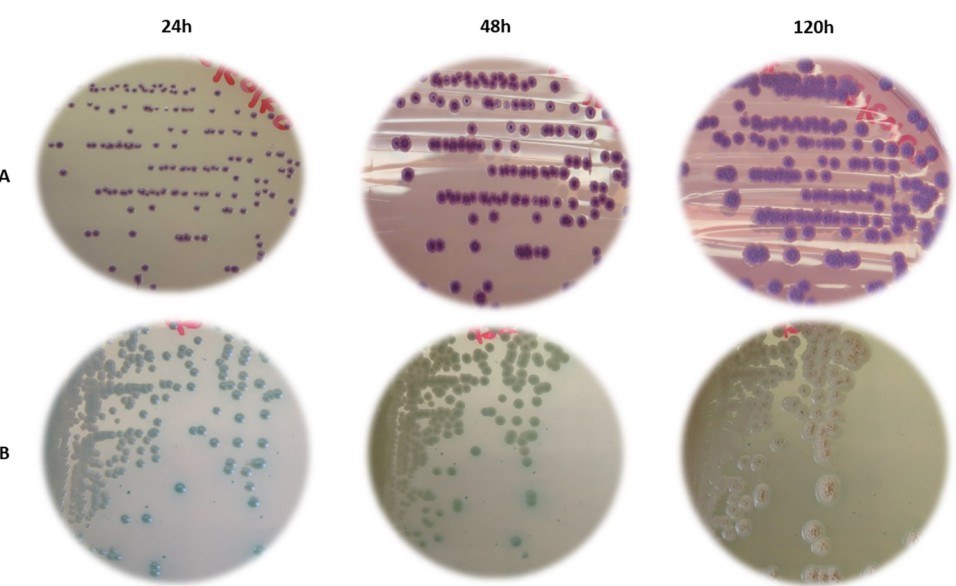

**Fig 4. Cultures of *Burkholderia pseudomallei* strain 22–10884_313#20.** Photographs of culture plates (A) on Ashdown agar and (B) on modified *B. cepacia* CHROMagar agar (supplemented with 4% glycerol, 500 mg/mL gentamicin and 130 mg/mL fosfomycin) after a 24, 48 and 120 h incubation at 37°C. The strain 22–10884_313#20 was isolated from soil 22–10884_313 collected from farm B in Les Saintes.

ST92 is identified in nine human strains: one from a Swiss traveler to Martinique [25], one from Puerto Rico, two from Mexico [42], and five from Brazil [43]. An MLST-based tree, including all STs identified so far in South America is shown in Fig 5.

## Discussion

Melioidosis has been extensively studied in established endemic regions such as Southeast Asia and Australia, with limited research in Central and South America [28,42]. In the Caribbean,

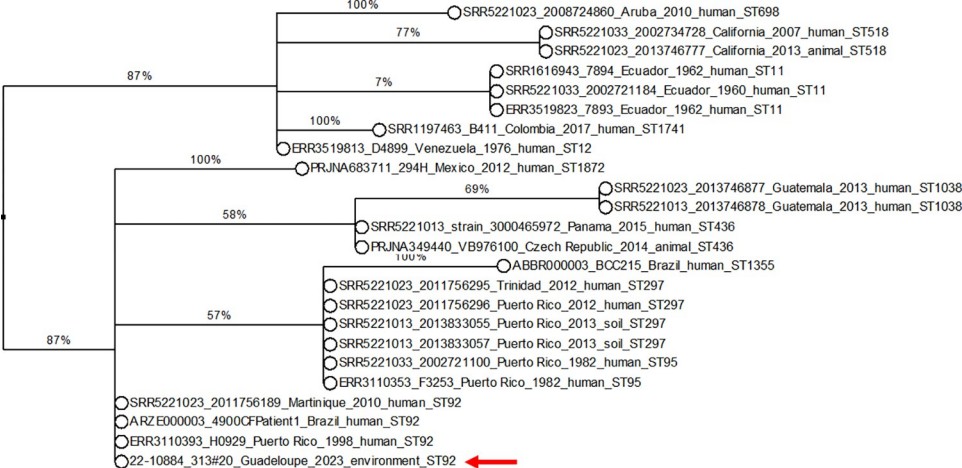

**Fig 5. MLST-based phylogenetic tree of 24 *B. pseudomallei* strains from South America, including the strain 22–10884_313#20 isolated in this study.** The phylogenetic tree was constructed from 24 South American strains: 23 from the PubMLST database and the strain isolated in this study (22–10884_313#20) indicated by a red arrow, which was isolated from a goat farm in Les Saintes. The MLST tree was a maximum parsimony tree with bootstrapping (n = 100). The numbers on the branches correspond to the bootstrap values.

human cases have been documented in Guadeloupe and Martinique for over 30 years [25–29], but the local presence of *B. pseudomallei* in these areas remains unconfirmed. Isolation of strains from the environment is challenging, probably more so in regions of emergence or low endemicity. This difficulty was highlighted in a recent environmental survey in Puerto Rico, a Caribbean island, where only three out of 500 soil samples from 60 sites tested positive for *B. pseudomallei* by PCR, and only one sample was positive by culture [44].

Previous exposure to *B. pseudomallei* triggers an immune response that can be detected by serological methods [45]. While serology isn't usually favored for melioidosis epidemiology in endemic areas due to suspected frequent environmental exposure and to the challenge of interpreting positive results outside of a clinical context [46,47], serological testing shows promise in areas of emergence. In particular, for animals confined to a restricted geographical area, serological testing allows for targeted environmental investigations within the living environment of positive animals, maximizing the chance of detecting the bacterium.

In our study, we used the GLANDA ELISA, originally developed for the diagnosis of equine glanders. We extended this method to the detection of antibodies against *B. pseudomallei* because a previous study carried out on a goat farm in New Caledonia, where a case of melioidosis was confirmed, demonstrated the efficacy of this kit in identifying animals exposed to *B. pseudomallei* [45]. Moreover, its dual antigen technology allows cross-species use without the need for species-specific conjugates. Our serological screening in French Guiana and Les Saintes showed that not all farms tested had seropositive animals, regardless of the species tested. This suggests that individual serological response may vary. In French Guiana, 2% of sheep and goats, 15% of cattle and 22% of horses tested positive, while in Les Saintes, where only goats were tested, 39% tested positive. In comparison, reported prevalences in endemic countries using other serological methods vary, such as 6% and 13.6% in sheep, 0.3% and 2.6% in goats, and 2% and 7.6% in cattle in Thailand [47] and Malaysia [46], respectively. However, differences in melioidosis endemicity and serological testing methods between countries make direct comparisons difficult. Also, our study included only two goat herds in Les Saintes and used opportunistic sampling in French Guiana, a sampling method that is not ideal to estimate prevalences.

In order to confirm our serology results, we carried out surveillance on two goat farms with seropositive animals in Les Saintes. Molecular screening in conjunction with an environmental survey confirmed fecal shedding of *B. pseudomallei* on one farm and successfully led to the isolation of an environmental strain of *B. pseudomallei* on the other farm. This confirms the usefulness of the GLANDA ELISA kit in animals to assess possible past environmental exposure to *B. pseudomallei*. However, due to the possibility of antibody cross-reactivity [48,49], direct detection of the bacterium is needed for conclusive evidence.

There is limited data on *B. pseudomallei* concentrations in the environment, but it's likely that the bacteria are present at low concentrations in areas of emergence or low endemicity, making detection more difficult. To increase the chance of detection, we optimized the protocol from existing guidelines [22] by testing two new media in addition to the conventionally recommended Ashdown medium [33]. First, an erythritol-based medium was used as the sole carbon source in the second step of a two-step enrichment to select for *B. pseudomallei* [23]. A similar two-step enrichment protocol (TBSS-C50 + erythritol) was recently used to isolate environmental *B. pseudomallei* strains in Ghana, West Africa [24]. Second, we used a chromogenic medium CHROMagar *B. cepacia* originally developed for the detection of *B. cepacia* in clinical samples [50]. To increase its selectivity for *B. pseudomallei*, the medium was modified by incorporating new antibiotics. We confirmed that reference strains of *B. pseudomallei* grew well on this modified medium (S1 Fig). To our knowledge, this is the first use of a chromogenic medium to improve the selection of *B. pseudomallei* colonies from poly-contaminated

environmental matrices. Despite these protocol improvements, in our study only one soil sample (out of 92 samples collected from seropositive farms) was found positive for *B. pseudomallei* by PCR, with a single strain isolated. Although the use of CHROMagar *B. cepacia* media appears promising, further protocol optimization (e.g., sample volume, enrichment time, chromogenic and/or selective medium for environmental matrices) will be required to facilitate the detection and isolation of environmental strains of *B. pseudomallei*. Of note, an additional soil sample was PCR positive for the *B. pseudomallei* complex but not for *B. pseudomallei* or *B. thailandensis* species. Further characterization of all non-*pseudomallei Burkholderia spp.* strains isolated in this study will help identify all the environmental *Burkholderia* species to which animals may be exposed and to assess their cross-reactivity with the GLANDA ELISA test.

Our study provides the first definitive confirmation of the local establishment of *B. pseudomallei* in Les Saintes, where two cases of human melioidosis were reported in 1997 (a tourist visiting Les Saintes for 3 weeks) [29] and in 2017 (a local resident) [27]. The MLST profile of the strain isolated in our study (ST92) was identical to that of the 1997 human case [29]. In the *Burkholderia pseudomallei* PubMLST database, ST92 was identified in nine human strains: one from a Swiss traveler to Martinique [25], one from Puerto Rico, two from Mexico [8] and five from Brazil [7], illustrating the spatial clustering of this sequence type.

Our study is subject to several limitations related to the use of the GLANDA ELISA test. A major uncertainty was the identity of the recombinant protein used in this commercial kit that has been validated for the diagnosis of equine glanders. Although we speculate that this undisclosed protein is also present in *B. pseudomallei* (due to the very close relationship between *B. mallei* and *B. pseudomallei*), it is possible that the antibodies detected by the test could be triggered by other environmental *Burkholderia* species, particularly those belonging to the *B. pseudomallei* complex. The presence of *B. thailandensis* in Les Saintes and French Guiana is uncertain, although it has recently been detected in the United States [48]. Therefore, the results obtained with this GLANDA ELISA kit are only an indication of the possible exposure of the animals to *B. pseudomallei*. In areas of emergence or low endemicity with little to no clinical cases, only the isolation of the bacterium can confirm the serological results.

Another limitation is related to equids and glanders. The distinction between glanders (caused by *B. mallei*) and melioidosis (caused by *B. pseudomallei*) is difficult due to their high antigenic similarity [17], with *B. mallei* being a monophyletic clade within *B. pseudomallei* [49]. As glanders is frequently reported in Brazil [51], a country bordering French Guiana, the detection of GLANDA ELISA positive horses in French Guiana warranted further investigation. Clinically healthy horses that tested GLANDA ELISA positive were all Hcp1 positive by Luminex but GroEL negative by Luminex, both proteins being key markers for the diagnosis of glanders [32]. When CFT (the reference method for the diagnosis of glanders) was added as a fourth test, all GLANDA ELISA positive horses were CFT negative, with the exception of horses on farm #24. On this farm, the detection of horses positive to three of the four tests relevant to the diagnosis of glanders led to a follow-up examination three months later, which confirmed the initial serological results and the absence of clinical signs typical of acute glanders (e.g., respiratory distress or abscesses). Despite the absence of previously reported cases of glanders or melioidosis in French Guiana, the ultimate health status of these equines remains uncertain given the chronic potential of these two diseases [35,52,53].

In conclusion, our study supports the hypothesis that animals could be used as sentinels to detect the presence of *B. pseudomallei* in the environment, with the GLANDA ELISA test being a relevant tool for this purpose in areas of emergence or where the endemicity of melioidosis is uncertain. However, definitive proof must be obtained by direct detection of the bacteria. Further research is also needed to establish the specificity of this test in contexts other than

those in which it was originally designed and validated. In addition, the threshold used in our study was originally established for equine glanders in a clinical setting and may require re-evaluation to assess environmental exposure to *B. pseudomallei* in different host species. Such refinement may be particularly useful in areas of emergence or low endemicity. This study confirmed the presence of the bacterium in Les Saintes. Similar serological and environmental studies now need to be carried out in Guadeloupe and Martinique, where human cases have also been reported [25–30]. Although no cases of melioidosis have been reported in French Guiana, our serological results suggest the presence of *B. pseudomallei* and the need for environmental surveys around the ELISA-positive farms identified in this study and for retrospective re(evaluation) of human infections of unconfirmed ethiology and with symptoms similar to those of melioidosis.

## Supporting information

**S1 Table. Serological results of animals in French Guiana and Les Saintes (Guadeloupe).**
(XLSX)

**S2 Table. Antibiogram results of 22–10884_313#20 strain.**
(XLSX)

**S1 Fig. Evaluation of CHROMagar *B. cepacia* as a relevant media for other *Burkholderia* strains.**
(TIF)

## Acknowledgments

We thank all the livestock farmers, especially the owners of the two goat farms in Les Saintes, who allowed repeated sampling to complete the longitudinal study. Special thanks to CHRO-Magar for supplying the culture media and to Dr. Sophie Granier for her valuable advice on the implementation of the antibiogram test. We thank Carina Hall and Nathan Stone for helpful early discussions regarding environmental samplings.

## Author Contributions

**Conceptualization:** Vanina Guernier-Cambert, Sébastien Breurec, Emma Rochelle-Newall, Karine Laroucau.

**Investigation:** Mégane Gasqué, Gil Manuel, Rachid Aaziz, Jules Terret, Thomas Deshayes.

**Methodology:** Mégane Gasqué, Karine Laroucau.

**Resources:** Gil Manuel, Xavier Baudrimont, Sébastien Breurec.

**Supervision:** Vanina Guernier-Cambert, Emma Rochelle-Newall, Karine Laroucau.

**Writing – original draft:** Mégane Gasqué, Karine Laroucau.

**Writing – review & editing:** Mégane Gasqué, Vanina Guernier-Cambert, Gil Manuel, Rachid Aaziz, Jules Terret, Thomas Deshayes, Xavier Baudrimont, Sébastien Breurec, Emma Rochelle-Newall, Karine Laroucau.

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
