## [Decision Letter · Decision Letter 0]

9 Apr 2024

Dear Mrs Guernier-Cambert,

Thank you very much for submitting your manuscript "Serological screening in animals combined with environmental surveys provides definite proof of the local establishment of *Burkholderia pseudomallei* in Guadeloupe" for consideration at PLOS Neglected Tropical Diseases. As with all papers reviewed by the journal, your manuscript was reviewed by members of the editorial board and by several independent reviewers. In light of the reviews (below this email), we would like to invite the resubmission of a significantly-revised version that takes into account the reviewers' comments. 

The reviewers agreed that the manuscript describes important research on Burkholderia pseudomallei. However they highlight several key concerns that must be addressed on review. First, there is need for greater clarity in the methods - as the current presentation left the reviewers confused about sample collection, and analytical methods. Strive to write methods in a way such that they could be replicated by others. Second, there were concerns about ethical approval for the work with animals, and that samples may have been collected before approval was obtained. Third, there is a request for careful editing of the manuscript to make sure the methods, results and interpretation are clear.

We cannot make any decision about publication until we have seen the revised manuscript and your response to the reviewers' comments. Your revised manuscript is also likely to be sent to reviewers for further evaluation.

Sincerely,

Elizabeth J Carlton

Section Editor

Stuart Blacksell

Section Editor

The reviewers agreed that the manuscript describes important research on Burkholderia pseudomallei. However they highlight several key concerns that must be addressed on review. First, there is need for greater clarity in the methods - as the current presentation left the reviewers confused about sample collection and analytical methods. Strive to write methods in a way such that they could be replicated by others. Second, there were concerns about ethical approval for the work with animals, and that, at samples may have been collected before approval was obtained. This should be clarified. Third, there is a request for careful editing of the manuscript to make sure the methods, results and interpretation are clear. Additional comments are provided by the reviewers, below.

Reviewer's Responses to Questions

**Key Review Criteria Required for Acceptance?**

**Methods**

-Are the objectives of the study clearly articulated with a clear testable hypothesis stated?

-Is the study design appropriate to address the stated objectives?

-Is the population clearly described and appropriate for the hypothesis being tested?

-Is the sample size sufficient to ensure adequate power to address the hypothesis being tested?

-Were correct statistical analysis used to support conclusions?

-Are there concerns about ethical or regulatory requirements being met?

Reviewer #1: -Are the objectives of the study clearly articulated with a clear testable hypothesis stated?

Answer: Yes

-Is the study design appropriate to address the stated objectives?

Answer: Unclear. It is unclear whether the study was designed to perform or not perform environmental sampling, longitudinal study and rectal swab in French Guana from the start. If the seropositivity was observed in French Guana, why those studies were not conducted in French Guana. 

It is also unclear how positive-control for sera for each species were from?. This is very important as the analysis for positivity and S/P% are exclusively based on how positive control for sera from each species are from. 

-Is the population clearly described and appropriate for the hypothesis being tested?

Answer: Unclear. The farm selection for the cross-sectional and longitudinal study is unclear. Convenience sampling is not wrong, but clarity and real reasons how those two were selected are needed. 

-Is the sample size sufficient to ensure adequate power to address the hypothesis being tested?

Answer: Unclear. Sample size calculation is not presented. 

-Were correct statistical analysis used to support conclusions?

Answer: Not relevant. Statistical tests were not used and p values are not presented. This is fine, as descriptive analysis alone in this study is fine. 

-Are there concerns about ethical or regulatory requirements being met?

Answer: No, there are no concerns. However, it is good to present the study approval no. from the relevant EC/IRB.

Reviewer #2: (No Response)

Reviewer #3: - Objectives are clearly articulated with a clear hypothesis

- Study design is appropriate, however some methods needs some more attention

- Statistical analysis looks corrects

- It is not sure how ethical clearance was obtained for the animal and soil sampling studies, there is no mention of ethical clearance

**Results**

-Does the analysis presented match the analysis plan?

-Are the results clearly and completely presented?

-Are the figures (Tables, Images) of sufficient quality for clarity?

Reviewer #1: -Does the analysis presented match the analysis plan?

Answer. Yes. 

-Are the results clearly and completely presented?

Answer. Unclear. It is difficult to read in many parts in both abstract and main text. Result is abstract is unclear whether all parts of the study has been done in two regions (Guadeloupe and French Guiana) similarly or not. 

-Are the figures (Tables, Images) of sufficient quality for clarity?

Answer. Table 2 and 4 should move to supplementary Tables. Improve visualization for Table 2 and 4 are recommended, that would be better than providing all raw data as the current Table 2 and 4.

Reviewer #2: (No Response)

Reviewer #3: - please see my reviewer comments, some analysis needs more attention

- Some results needs more attention - see my reviewer comments

- Figures need more attention are not from sufficient quality/clarity

**Conclusions**

-Are the conclusions supported by the data presented?

-Are the limitations of analysis clearly described?

-Do the authors discuss how these data can be helpful to advance our understanding of the topic under study?

-Is public health relevance addressed?

Reviewer #1: -Are the conclusions supported by the data presented?

Answer: Yes. 

However,

(1) the abstract is not clearly written. 

(1.1) For example, "the highest prevalence rates were observed ..." , it is unclear of what this prevalence is; Seroprevalence by ELISA? Soil, water, goat rectal swab prevalence, ? 

(1.2) The abstract is not clearly written that the environmental sampling was not performed in French Guiana - all numerators and denominators of all testing in those two areas (Guadeloupe and French Guiana) for serological tests and environmental sampling should be clearly written. 

(2) The Author summary is not clearly written. 

English in many parts of this manuscript need intensive editing. 

(2.1) The key thing about science is that this approach (serological sampling + environmental sampling) should be considered in "new" areas or areas where endemicity of melioidosis is unclear. This method is not meaningful in known melioidosis-endemic areas. This point is unclear in the final sentence/conclusion of the author summary, "thereby helping to identify areas where the pathogen may be present in the environment."

-Are the limitations of analysis clearly described?

Answer: No. The limitation of accuracy, sample size, positive control, bias are not well written. 

-Do the authors discuss how these data can be helpful to advance our understanding of the topic under study?

Answer: Yes.

-Is public health relevance addressed?

Answer: Yes.

Reviewer #2: (No Response)

Reviewer #3: - the main conclusions are supported by the data presented

- limitations of the study are not clearly described

- Yes

- Yes

**Editorial and Data Presentation Modifications?**

Reviewer #1: (No Response)

Reviewer #2: Considerations for authors:

#63, 64, 65. We aimed to document animal exposure through a serology study, to document their immune response over time, and to investigate the link between seropositive animals and the presence of B. pseudomallei in their close environment.

Comment: It is important to draw attention to the interpretation of the serological test. The authors should not state the positivity of the test for B. pseudomallei in any part of the work since the ELISA kit was developed for diagnostic of glanders. Seropositivity for Bp is "possible", but not definitive.

Confusing text, especially in terms of method, due to the mixing of parts of the studies (serology, environment and laboratory study of Burkholderiapseudomallei).

Suggestion: organization of the items from the method. For example, serological studies (survey, additional serology) followed by environmental study and sample processing/bacteria characterization. It is also recommended that the result and discussion follow the same sequence.

# 103. Animal sampling protocols were approved by the IRD Ethics Committee (21 April 2022)

# 85. Blood samples were collected between November 2021 and December 2022.

Is this information correct? Were samples collected before ethical approval?

# 100. Rectal swabs were collected in duplicate for each animal between April 2022 and 101 December 2022.

Inform that the swab collection was parallel to the blood sample collection (in methods).

Reviewer #3: (No Response)

**Summary and General Comments**

Reviewer #1: This study would have some value for the community if it's well written and clear for all parts involved.

Reviewer #2: The manuscript discusses the presence of melioidosis in Guadeloupe and possible occurrence in French Guiana, raising the important debate about the distribution of the disease beyond the areas considered endemic. I also highlight the approach involving the environment and animal studies, fundamental for the complex ecological understanding of melioidosis. The diversity of Burkholderia species and the need for attention to the development of more specific diagnostic tests so that we can have a better understanding of diseases that affect animals such as melioidosis and glanders, was also considered in the study.

Reviewer #3: Reviewer’s comments PNTD-D-24-00204

This article aimes to investigate the presence of B. pseudomallei in animals and environment in

French West. The article is important to increase awareness of B. pseudomallei outside the known 

endemic hot spots. 

The authors are to be applauded for their efforts to put the melioidosis problem in French West on 

the map. Indeed, more understanding of the epidemiology is urgently needed because of the high

burden and the neglected status of melioidosis. However, the article needs more attention and the 

English and figures/tables should be further improved.

Major comments:

- Please remove ‘definite proof’ from the title and change in for example presence etc. As 

the serological study does not provides definite proof. Be aware that with this title no 

reader will understand you also performed a study in French Guiana.

- To me, Figure 2 together with the methods section at line 95-103, in which the authors refer 

to the serological analyses being explained later, is quite confusing. Also, the Les Saintes 

farms contain 31 animals (line 88) which contradicts with the numbers in Figure 2 as these 

are oftentimes higher dan 31. If both farms have 31 animals, please state this. The figure 

should be self-explanatory. 

- Explain how water samples were collected in the paragraph “Environmental sampling in 

Guadeloupe”. This information is absent. For example, how much water was collected, how 

was it stored, etcetera. 

- Please elaborate on the ethical approval for the study, how was ethical clearance obtained 

for the animal and soil sampling studies?

- Line 200: why was the supernatant used for PCR and not the heat-killed bacterial pallet?

- Line 205: why were gentamycin and colistin not tested in the antibiotic disc test?

- Line 212: why was a B. pseudomallei strain for example 1026 b or K96 not used as reference 

control?

- Line 223: it would be interesting to use whole genome sequencing on the obtained B. 

pseudomallei strains

- Line 247: the following sentence is obsolete “The serological results showed heterogeneity 

both between farms, differentiating 247 positive and negative farms, as well as within 

positive farms, where both positive and negative animals were detected”. You will always 

have positive and negative findings within farms and within animals so this does not explain 

heterogeneity in the results. Please remove this sentence. 

- In Figure 3, several remarks have to be made as the figure is not clear 

o Please state where the sheep and equids originate from in the legend.

o Panel C is too small. 

o The numbers in blue are frequently incorrectly aligned. Also, I am not sure if this 

should be displayed here. It might be better to mention this together with the farm 

IDs and place the median S/P rate on top of the error bars. 

o In the legend, please first list all the farms in French Guiana and then in 

Guadeloupe. Also, rearrange the panels in a similar way. 

- Table 2 should be rearranged and preferably removed from the main text. Please pay 

attention to the following:

o Unsure what the added value is of presenting the data per sampled equid. This can 

also be summarized in the text. 

o The distinction between the farms is unclear. Might be better to use column 1 for 

this. 

o Results should not be mentioned in footnotes.

o Tables should be self-explanatory: state why sometimes two results are shown per 

study ID, such as “P / P” or “ns / N”

- Table 3 requires some changes:

o First of all, I was under the impression only 31 animals were present on the Les

Saintes farms. However, on December 2022 on Farm A, a total of 39 animals were 

tested. Please also see my comments regarding Figure 2. 

o Please remove the CT values. You either consider a case positive or not. 

- Table 4 should be rearranged. I think the data can easily be summarized in a smaller table in 

which a comparison is made between farm A and B. For example, state the number of 

males/females, the age with appropriate statistics based on the distribution of the data, 

and the pooled positivity rates with, for example, a range of the data. 

- Line 332-333: I am not sure how the sample IDs contribute here. Suggest to remove them. 

This applies to the whole text. 

- Table 5 should be rearranged. Please pay attention to the following:

o Please remove the CT values. You either consider a case positive or not. 

- The sequence data of strain D22-10884_S313#20 should be uploaded to an online repository. 

For example, the ENA database. 

- Please include a phylogenetic tree that compares the MLST profile of your identified strain 

to the existing B. pseudomallei strains, especially from South America. 

- Please mention the limitations of the study clearly in a separate paragraph and some of the 

discussion could be condensed. (For example the serological method used in this paper and 

the cross reactivity to B. mallei (and B. thailandensis?). What kind of serological methods 

would the authors preferable use in future studies?

Minor comments:

- Abstract: please note that the reader won’t know were Les saintes is (include Guadeloupe in 

the conclusion)

- Line 3: sentence should not start with ‘If…’. Please rephrase. 

- Line 4: please split sentence in 2. Recent studies ..

- Line 9: please change the pathogen to B. pseudomallei

- Line 10: please include serological à combined serological surveys

- Line 11: without clinical signs of melioidosis?

- Line 19: (B): is unnecessary please remove this

- Line 19: please use another word for ‘telluric’. This is not commonly used. 

- Line 10: Change it to B. pseudomallei

- Line 27: references 3-5 don’t refer to presence of B. pseudomallei in Africa. 

- Line 31: Please change Several risk factors to Risk factors, such as …

- Line 37: It is not clear where these refers to.

- Line 51-53: the following sentence is incorrect “Its detection could be harder in nonendemic areas where the bacterium may be present at lower levels”. This sentence 

contradicts itself as a bacterium cannot be isolated from the soil if the bacterium is nonendemic. Please rephrase. 

- Line 58: Some of whom had no history of travel to known endemic areas

- In Figure 1, panel A it is very hard to see the geographical distribution of the sampling 

sights. Please add an inset of Les Saintes with all these sites distinguishable from each 

other. 

- In Figure 2 the resolution appears to be low. Please add a high quality figure. 

- Line 14-15 is it 5m or 6m appart?

- Line 149: explain MRI#1 as the reader will not familiar with this abbreviation 

- Line 234: it will not be clear to the reader what you mean with S/P%, please explain

- Line 292: what does complex means here?

- Line 293: remove the word “species”

- Line 346: please do not use the following abbreviation CHR_GGF

- Line 346: suspected colonies? Please remove this whole sentence as this is not of interest. 

Same applies to sentence 348 Fifteeen suspect coloncies were isolated from soil .. For 

example you can start instead with: ‘Fifteen colonies were tested PCR-positive for the B. 

pseudomallei complex but ..’ etc

- Line 363: why was only one colony selected for further analysis?

- Line 381: please explain what you mean with CFT

- Line 383: explain what you mean with “within-farm variability”

- Line 387-388: the authors mention “within-farm variability” but they give an explanation for 

between-farm variability “why animals living in farms with similar environmental exposure 

do not all show the same serological result”. Rephrase this so it is correctly explained.

PLOS authors have the option to publish the peer review history of their article (what does this mean?). If published, this will include your full peer review and any attached files.

Reviewer #1: No

Reviewer #2: Yes: Dionne B Rolim

Reviewer #3: No
---

## [Decision Letter · Decision Letter 1]

28 Aug 2024

Dear Mrs Guernier-Cambert,

We are pleased to inform you that your manuscript 'Reassessing the distribution of *Burkholderia pseudomallei* outside known endemic areas using animal serological screening combined with environmental surveys: the case of Les Saintes (Guadeloupe) and French Guiana.' has been provisionally accepted for publication in PLOS Neglected Tropical Diseases.

Best regards,

Elizabeth Carlton

Section Editor

Reviewer's Responses to Questions

**Key Review Criteria Required for Acceptance?**

**Methods**

-Are the objectives of the study clearly articulated with a clear testable hypothesis stated?

-Is the study design appropriate to address the stated objectives?

-Is the population clearly described and appropriate for the hypothesis being tested?

-Is the sample size sufficient to ensure adequate power to address the hypothesis being tested?

-Were correct statistical analysis used to support conclusions?

-Are there concerns about ethical or regulatory requirements being met?

Reviewer #1: (No Response)

**Results**

-Does the analysis presented match the analysis plan?

-Are the results clearly and completely presented?

-Are the figures (Tables, Images) of sufficient quality for clarity?

Reviewer #1: (No Response)

**Conclusions**

-Are the conclusions supported by the data presented?

-Are the limitations of analysis clearly described?

-Do the authors discuss how these data can be helpful to advance our understanding of the topic under study?

-Is public health relevance addressed?

Reviewer #1: (No Response)

**Editorial and Data Presentation Modifications?**

Reviewer #1: (No Response)

**Summary and General Comments**

Reviewer #1: The authors have revised the manuscript intensively and I have no further major concerns.

PLOS authors have the option to publish the peer review history of their article (what does this mean?). If published, this will include your full peer review and any attached files.

Reviewer #1: No

---

## [Editor Report · Acceptance letter]

20 Sep 2024

Dear Mrs Guernier-Cambert,

We are delighted to inform you that your manuscript, "Reassessing the distribution of *Burkholderia pseudomallei* outside known endemic areas using animal serological screening combined with environmental surveys: the case of Les Saintes (Guadeloupe) and French Guiana.," has been formally accepted for publication in PLOS Neglected Tropical Diseases.

Best regards,

Shaden Kamhawi

co-Editor-in-Chief

Paul Brindley

co-Editor-in-Chief
